# BISIMULATION METRIC FOR MODEL PREDICTIVE CONTROL

**Yutaka Shimizu & Masayoshi Tomizuka**
Mechanical Engineering
University of California, Berkeley
Berkeley, CA 94720, USA
`{purewater0901, tomizuka}`@berkeley.edu

## ABSTRACT

Model-based reinforcement learning has shown promise for improving sample efficiency and decision-making in complex environments. However, existing methods face challenges in training stability, robustness to noise, and computational efficiency. In this paper, we propose Bisimulation Metric for Model Predictive Control (BS-MPC), a novel approach that incorporates bisimulation metric loss in its objective function to directly optimize the encoder. This time-step-wise direct optimization enables the learned encoder to extract intrinsic information from the original state space while discarding irrelevant details and preventing the gradients and errors from diverging. BS-MPC improves training stability, robustness against input noise, and computational efficiency by reducing training time. We evaluate BS-MPC on both continuous control and image-based tasks from the DeepMind Control Suite, demonstrating superior performance and robustness compared to state-of-the-art baseline methods.

## 1 INTRODUCTION

Reinforcement learning (RL) has become a central framework for solving complex sequential decision-making problems in diverse fields such as robotics, autonomous driving, and game playing. Among RL methods, model-based reinforcement learning (MBRL) gets its attention thanks to its ability to achieve higher sample efficiency and better generalization. Representation learning further enhances MBRL by encoding high-dimensional information into compact latent spaces, which can accelerate learning by focusing on essential aspects of the environment. However, achieving stable and robust representations remains a challenge, especially in high-dimensional or partially observable environments, where noise and irrelevant features can degrade performance.

One prominent MBRL method, Temporal Difference Model Predictive Control (TD-MPC) (Hansen et al., 2022), combines temporal difference learning with model predictive control to improve policy performance by simulating future trajectories in the learned latent space. TD-MPC sets itself apart from other methods by leveraging the learned latent value function as a long-term reward estimate to approximate cumulative rewards, allowing for the efficient computation of optimal actions. Despite its successes, TD-MPC suffers from several limitations, including instability during training, vulnerability to noise, and expensive computational costs, which are shown in Fig 1. The first graph illustrates TD-MPC's performance degradation during training, demonstrating a notable collapse after a certain number of steps. The second set of results focuses on an image-based task, where the addition of background noise (adding a completely irrelevant image to the background) led to TD-MPC's failure to achieve a high reward in the noisy environment. The third picture shows that TD-MPC suffers from a long calculation time. These problems are attributed to the encoder's training method and the objective function's structure.

To address these issues, we introduce Bisimulation Metric for Model Predictive Control (BS-MPC), a new approach that leverages $\pi^*$-bisimulation metric (on-policy bisimulation metric) (Zhang et al., 2021) to improve the stability and robustness of latent space representations. Bisimulation metrics measure behavioral equivalence between states by comparing their immediate rewards and next state distributions, providing a formal way to ensure that the learned latent representations retain

meaningful and essential information from the original states. In BS-MPC, we minimize the mean square error between the on-policy bisimulation metric and $\ell_1$-distance in latent space at each time step, directly optimizing the encoder to improve stability and noise resistance. Integration of the bisimulation metric gives BS-MPC a theoretical guarantee, ensuring that the difference in cumulative rewards between the original state space and the learned latent space can be upper-bounded over a trajectory. This value function difference bound validates the fidelity of the encoder projection. Additionally, the proposed method reduces training time by making the computation of the objective function parallelizable, leading to less computational cost than TD-MPC. All these performance improvements are also summarized in Fig 1.

We implement BS-MPC using the Model Predictive Path Integral (MPPI) (Williams et al., 2016; 2018) framework and evaluate its performance on a variety of continuous control tasks from Deep-Mind Control Suite (Tassa et al., 2018). Our results show that BS-MPC outperforms existing model-free and model-based methods in terms of performance and robustness, making it a promising new approach for model-based reinforcement learning.

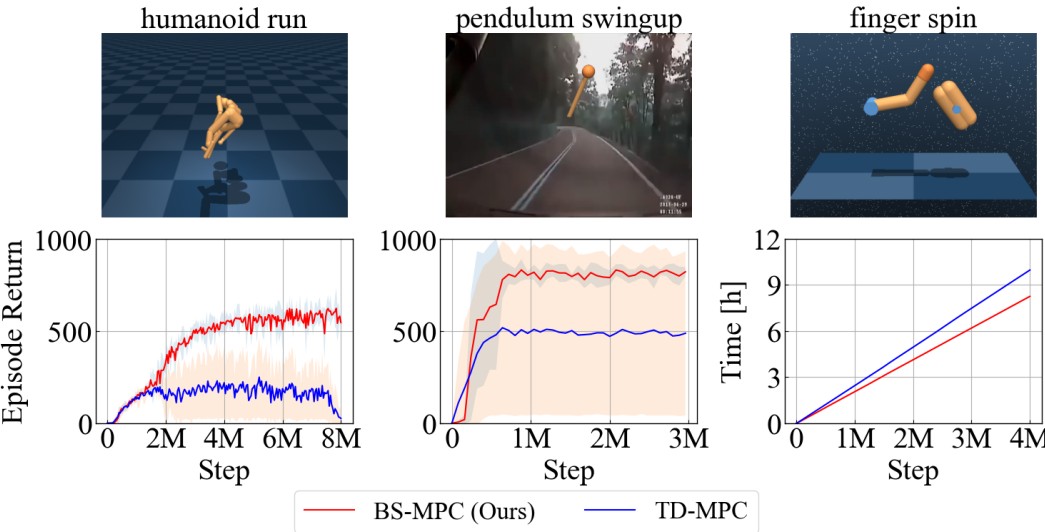

**Figure 1:** Three open problems of TD-MPC. (Left) TD-MPC initially performs well but collapses after 4 million steps, while BS-MPC steadily improves. (Middle) With added distraction in the input image, TD-MPC fails to gain rewards, whereas BS-MPC remains robust. (Right) BS-MPC reduces training time by removing sequential computation in objective function.

## 2 RELATED WORK

**Reinforcement Learning**   Reinforcement Learning (RL) (Sutton & Barto, 2018) has two main approaches: model-free methods (Silver et al., 2014; Fujimoto et al., 2018; Haarnoja et al., 2018a; Schulman et al., 2015; 2017; Kalashnkov et al., 2021; Kalashnikov et al., 2018; Mnih et al., 2015; Hessel et al., 2018; Yarats et al., 2021; 2022; Laskin et al., 2020) and model-based methods (Sutton, 1991; Hafner et al., 2020; 2021; 2024; Luo et al., 2019; Janner et al., 2019; Chua et al., 2018; Schrittwieser et al., 2019; Wang & Ba, 2020). While model-free methods focus on learning the value function and policy, model-based methods aim to learn the underlying model of the environment, using this learned model to compute optimal actions. This paper focuses on the model-based approach, specifically methods that combine planning and MBRL (Hansen et al., 2022; 2024), which learns the underlying model in the latent space and applies sampling-based Model Predictive Control (MPC) (Williams et al., 2016; Kobilarov, 2012) to solve the trajectory optimization problem. Several variants of TD-MPC have been proposed (Lancaster et al., 2024; Zhao et al., 2023; Chitnis et al., 2023; Feng et al., 2023; Wan et al., 2024), but none fully address all the challenges outlined in Fig. 1. To the best of our knowledge, BS-MPC is the first approach to tackle all three open problems in TD-MPC, as discussed in Section 1.

**Representation Learning**  Learning models in the latent space is an efficient way to approximate internal models, especially for image-based tasks. One approach to learning latent space projections is by training both the encoder and decoder to minimize the reconstruction loss (Lange & Riedmiller, 2010; Lange et al., 2012; Hafner et al., 2019; 2024; Lee et al., 2020). However, this method often suffers model errors and instability and has difficulties in long-term predictions. An alternative approach is to train only the encoder to obtain the latent representation. Bisimulation (Larsen & Skou, 1989) is a state abstraction technique defined in Markov Decision Processes (MDPs) that clusters states producing identical reward sequences for any given action sequence. Ferns et al. (2011); Ferns & Precup (2014) defined a bisimulation metric that measures the similarity between two states based on the Wasserstein distance between their empirically measured transition distributions. However, computing this metric can be computationally expensive in high-dimensional spaces. To address this, Castro (2020) proposed an on-policy bisimulation metric, which considers the distribution of future states under the current policy, providing a scalable way to measure state similarity. Zhang et al. (2021) extend this idea to $\pi^*$-bisimulation metric by minimizing MSE loss between bisimulation metric and $\ell_1$-distance in latent space to train the encoder. Following this approach, we use the on-policy bisimulation metric to train the encoder in model-based reinforcement learning architecture.

## 3 PRELIMINARIES

This section provides a brief introduction to reinforcement learning and its associated notations, along with an explanation of bisimulation concepts.

### 3.1 REINFORCEMENT LEARNING

Reinforcement Learning (RL) aims to optimize agents that interact with a Markov Decision Process (MDP) defined by a tuple $(\mathcal{S}, \mathcal{A}, \mathcal{P}, \mathcal{R}, \rho_0, \gamma)$, where $\mathcal{S}$ represents the set of all possible states, $\mathcal{A}$ is the set of possible actions, $\mathcal{R}$ is a reward function, $\rho_0$ is the initial state distribution, and $\gamma$ is the discount factor. When action $a \in \mathcal{A}$ is executed at state $s \in \mathcal{S}$, the next state is generated according to $s' \sim \mathcal{P}(\cdot|s, a)$, and the agent receives stochastic reward with mean $r(s, a) \in \mathbb{R}$.

The Q-function $Q^\pi(s, a)$ for a policy $\pi(\cdot|s)$ represents the discounted long-term reward attained by executing $a$ given observation history $s$ and then following policy $\pi$ thereafter. $Q^\pi$ satisfies the Bellman recurrence: $Q^\pi(s, a) = \mathbb{B}^\pi Q^\pi(s, a) = r(s, a) + \gamma \mathbb{E}_{s' \sim P(\cdot|s,a), a' \sim \pi(\cdot|s')} [Q_{h+1}(s', a')]$ . The value function $V^\pi$ considers the expectation of the Q-function over the policy $V^\pi(h) = \mathbb{E}_{a \sim \pi(\cdot|s)} [Q^\pi(s, a)]$. Meanwhile, the Q-function of the optimal policy $Q^*$ satisfies: $Q^*(s, a) = r(s, a) + \gamma \mathbb{E}_{s' \sim P(\cdot|s,a)} [\max_{a'} Q^*(s', a')]$, and the optimal value function is $V^*(s) = \max_a Q^*(s, a)$. Finally, the expected cumulative reward is given by $J(\pi) = \mathbb{E}_{s_1 \sim \rho_1} [V^\pi(s_1)]$. The goal of RL is to optimize a policy $\pi(\cdot \mid s)$ that maximizes the cumulative reward $\pi^*(\cdot \mid s) = \operatorname{argmax}_\pi J(\pi)$.

In large-scale or continuous environments, solving reinforcement learning (RL) problems can be challenging due to the prohibitively high computational cost. To address this issue, function approximation is often employed to estimate value functions and policies. With function approximation, we present $Q^\pi, \pi$ as $Q_{\theta^Q}, \pi_\psi$, with $\theta^Q$ and $\psi$ as their parameters. With a replay buffer $\mathcal{B}$, the policy evaluation and improvement steps at iteration $k$ can be expressed as:

$$\begin{aligned} \theta_{k+1}^Q &\leftarrow \operatorname*{argmin}_{\theta^Q} \mathbb{E}_{(s,a,r,s') \sim \mathcal{B}} \left[ \left( Q_{\theta^Q}(s, a) - \mathcal{R}(s, a) - \gamma \mathbb{E}_{a' \sim \pi_{\theta_k^\pi}(\cdot|s)} [Q_{\bar{\theta}_k^Q}(s, a')] \right)^2 \right] \\ \psi_{k+1} &\leftarrow \operatorname*{argmax}_{\psi} \mathbb{E}_{s \sim \mathcal{B}, a \sim \pi_\psi(\cdot|s)} \left[ Q_{\theta_{k+1}^Q}(s, a) \right] , \end{aligned} \tag{1}$$

where $\bar{\theta}_k^Q$ are target parameters that are a slow-moving copy of $\theta_k^Q$. Note that in this paper, we denote $\bar{\bullet}$ as target parameters in this paper.

### 3.2 BISIMULATION METRIC

When working with high-dimensional state problems, it is often helpful to group "similar" states into the same set. Bisimulation is a type of state abstraction that groups state $s_i$ and $s_j$ if they are

"behaviorally equivalent" (Li et al., 2006). A more concise definition states that two states are bisimilar if they yield the same immediate rewards and have equivalent distributions over future bisimilar states(Larsen & Skou, 1989; Givan et al., 2003). Bisimulation metric quantifies the bisimilarity of two states $s_i$ and $s_j$. It is defined with $p$-th Wasserstein metric $W_p(\mathcal{P}_1, \mathcal{P}_2)$, which represents the distance between two probability distribution $\mathcal{P}_1$ and $\mathcal{P}_2$:

**Definition 1.** (Bisimulation metric (Ferns et al., 2011)). The following metric exists and is unique, given $R : S \times A \to [0, 1]$ and $c \in (0, 1)$ for continuous MDPs:

$$d(s_i, s_j) = \max_{a \in \mathcal{A}} (1 - c)|\mathcal{R}(s_i, a) - \mathcal{R}(s_j, a)| + cW_1(\mathcal{P}(\cdot|s_i, a), \mathcal{P}(\cdot|s_j, a)). \qquad (2)$$

In high-dimensional and continuous environments, analytically computing the max operation in Eq. 2 is challenging. In response to this difficulty, Castro (2020) proposed a new approach, known as the on-policy bisimulation metric (or $\pi$-bisimulation).

**Definition 2.** (On-Policy bisimulation metric (Castro, 2020)). Given a fixed policy $\pi$, the following bisimulation metric uniquely exists

$$d(s_i, s_j) = |r_{s_i}^\pi - r_{s_j}^\pi| + \gamma W_1(\mathcal{P}^\pi(\cdot|s_i), \mathcal{P}^\pi(\cdot|s_j)). \qquad (3)$$

where

$$r_s^\pi = \sum_a \pi(a|s)\mathcal{R}(s, a), \quad \mathcal{P}^\pi(\cdot|s) = \sum_a \pi(a|s) \sum_{s' \in S} \mathcal{P}(s'|s, a) \qquad (4)$$

Recently, (Zhang et al., 2021) extended the concept of the on-policy bisimulation metric (referred to as the $\pi^*$-bisimulation metric) to learn a comparable metric in the latent space $\mathcal{Z}$. In their approach, the encoder $\phi$ is learned by minimizing the mean square error between the on-policy bisimulation metric and $\ell_1$-distance in the latent space.

$$J(\phi) = \left( \|\phi(s_i) - \phi(s_j)\|_1 - |r_{s_i}^\pi - r_{s_j}^\pi| - \gamma W_2 \left( \hat{\mathcal{P}}(\cdot|\bar{\phi}(s_i), a_i), \hat{\mathcal{P}}(\cdot|\bar{\phi}(s_j), a_j) \right) \right)^2 \qquad (5)$$

where the latent dynamics model $\hat{\mathcal{P}}$ is modeled with a Gaussian distribution. In Eq. 5, 2-Wasserstein metric $W_2$ is used because it has a convenient closed form for Gaussian distribution. Following this approach, we train our encoder similarly by including this MSE loss (Eq. 5) in our objective function.

## 4 BISIMULATION METRIC FOR MODEL PREDICTIVE CONTROL

We introduce Bisimulation Metric Model Predictive Control (BS-MPC), a robust and efficient model-based reinforcement learning algorithm. This section provides a detailed explanation of the BS-MPC algorithm. Furthermore, we present a theoretical analysis that bounds the suboptimality of cumulative rewards in the learned latent space under our architecture. Finally, we highlight three key distinctions between BS-MPC and TD-MPC that contribute to their performance differences.

### 4.1 BISIMULATION METRIC FOR MODEL PREDICTIVE CONTROL

We introduce BS-MPC, an improvement over TD-MPC that employs $\pi^*$-bisimulation metrics to train the encoder. The training flow for BS-MPC is detailed in Appendix C.

**Components** BS-MPC shares the same five core components as TD-MPC: encoder, latent dynamics, reward, state-action value and policy.

$$
\begin{aligned}
&\textbf{Encoder:} \quad z_k = h_{\theta^h}(s_k) \qquad\qquad \textbf{Latent dynamics:} \quad z_{k+1} = d_{\theta^d}(z_k, a_k) \\
&\textbf{Reward:} \quad \hat{r}_k = \mathcal{R}_{\theta^R}(z_k, a_k) \qquad \textbf{State-action value:} \quad \hat{Q}_k = Q_{\theta^Q}(z_k, a_k) \qquad (6) \\
&\textbf{Policy:} \quad \hat{a}_k \sim \pi_\psi(z_k)
\end{aligned}
$$

When the input $s_k$ is a state vector, the encoder is modeled as a multi-layer perceptron (MLP) and as a convolutional neural network (CNN) when $s_k$ is an image. Given the latent state $z_k$ and action $a_k$, we compute the next latent state $z_{k+1}$ using the latent dynamics model $d_{\theta^d}(z_k, a_k)$, parameterized by $\theta^d$. Following other model-based reinforcement learning methods, we model the latent dynamics with an MLP. BS-MPC estimates the reward $\hat{r}_k$ and state-action value $\hat{Q}_k$ based on $z_k$ and $a_k$,

modeling both $\mathcal{R}_{\theta^{\mathcal{R}}}$ and $Q_{\theta^Q}$ with MLPs. Finally, we train a policy that outputs the estimated optimal action $\hat{a}_k$ given $z_k$; the policy is also parameterized as an MLP.

At each time step $k$, the original observation $s_k$ is encoded into the latent state $z_k$. Using $z_k$ and the action $a_k$, we compute the rewards, state-action values, and the next latent state. As highlighted in prior work, these values are computed in the latent space rather than the original observation space, as the latent state $z_k$ captures the essential information from the high-dimensional original state. Since the latent space typically has more compact dimensions, this approach is commonly used in image-based tasks where input images are high-dimensional. However, state-based tasks, despite being represented more compactly, also benefit from this structure by utilizing latent states learned through temporal consistency (Zhao et al., 2023).

**Objective function**  We jointly train the encoder, latent dynamics model, reward model, and state-action value model. BS-MPC minimizes the following loss function:

$$\theta^* = \arg\min_{\theta} \mathcal{L}(\theta) = \arg\min_{\theta} \; \mathbb{E}_{(s,a,r,s') \sim \mathcal{B}} \left[ \sum_{k=0}^{H} \lambda^k \mathcal{L}_k(\theta) \right] \tag{7}$$

where $\theta = [\theta^{\mathcal{R}}, \theta^Q, \theta^d, \theta^h]$. This objective function is identical to the one proposed in TD-MPC. However, BS-MPC has an additional bisimulation metric loss term at every time step, as shown in Eq. 5.

$$\mathcal{L}_k(\theta) = c_1 \underbrace{||\mathcal{R}_{\theta^{\mathcal{R}}}(z_k, a_k) - r_k||_2^2}_{\text{(A) reward loss}} + c_2 \underbrace{||Q_{\theta^Q}(z_k, a_k) - (r_k + \gamma Q_{\bar{\theta}^Q}(z_{k+1}, \pi_{\theta^\pi}(z_{k+1})))||_2^2}_{\text{(B) state-action value loss}}$$

$$+ c_3 \underbrace{||d_{\theta^d}(z_k, a_k) - h_{\bar{\theta}^h}(s_{k+1})||_2^2}_{\text{(C) consistency loss}} + c_4 \underbrace{\left( ||h_{\theta^h}(s_k) - h_{\theta^h}(\acute{s}_k)||_1 - |r_k - \acute{r}_k| - \gamma||\bar{z}_{k+1} - \bar{\acute{z}}_{k+1}||_2^2 \right)^2}_{\text{(D) bisimulation metric loss}}$$
$$\tag{8}$$

where $\acute{\bullet}_k = \text{permute}(\bullet_k)$ and $\bar{z}_{k+1} = d_{\bar{\theta}^d}(z_k, a_k)$. $c_1, c_2, c_3, c_4$ are parameters that can change the weight of each loss. The last term is an expansion of Eq. 5, under the assumption that the model outputs deterministic predictions, corresponding to a Dirac delta distribution (i.e., a Gaussian distribution with zero variance). As in TD-MPC, we use the same three loss components: (A) reward loss, (B) state-value action loss, and (C) consistency loss. Each training loss aims to update its corresponding parameters, i.e. reward parameter $\theta^{\mathcal{R}}$, state-action value parameter $\theta^Q$, and dynamics parameter $\theta^d$. These losses also help to update the encoder parameter $\theta^h$ by using the derivative of the composition function. In addition to these losses, BS-MPC includes a bisimulation metric loss in its objective function, which explicitly depends on the encoder parameters $\theta^h$. This bisimulation metric loss (D) is designed to train the encoder to learn a representation space where the $\ell_1$-distance corresponds to the $\pi^*$-bisimulation metric.

For policy training, we use the following loss function to update the policy parameter $\psi$.

$$\psi^* = \arg\min_{\psi} \mathcal{J}_\pi(\psi) = \arg\min_{\psi} \; -\sum_{k=0}^{H} \lambda^k Q_{\theta^Q}\left(z_k, \pi_\psi(\bar{z}_k)\right) \tag{9}$$

In Section. 4.3, we give details about the benefit of using Eq. 8 as the objective function for BS-MPC and the differences between our approach and TD-MPC.

**Model Predictive Control with Learned Model**  Following TD-MPC, our method has a closed-loop controller using the learned latent dynamics model, reward model, state-value function, and prior policy to compute the optimal action. Due to the high affinity between reinforcement learning and sampling-based planners, we design a closed-loop controller using MPPI, a type of sampling-based MPC, following the approach of TD-MPC. MPPI is a derivative-free method that samples a large number of trajectories, calculates the weight for each, and then generates the optimal trajectory by taking the weighted average of these trajectories. This framework enables us to solve the local trajectory optimization problem.

First, it encodes the current observed state $s_t$ into the latent space with the trained encoder $z_t = h_{\theta^h}(s_t)$. After that, we sample $M$ action sets from Gaussian distribution $\mathcal{N}(\mu^0, \sigma^0)$ based on the

initial mean $\mu^0$ and standard deviation $\sigma^0$, and each set contains $H$ length actions $a_{t:t+H}^j$ where $j \in M$. Starting from the initial latent state $z_t$, we use the learned latent dynamics $z_{t+1} = d_{\theta^d}(z_t, a_t)$ and sample $M$ trajectories. We then calculate the weight of each trajectory based on its cost and compute the weighted mean of the sampled trajectories to get the updated optimal trajectory. The parameter $\mu_k$ and $\sigma_k$ is updated by maximizing the following equations:

$$\mu^{k+1}, \sigma^{k+1} = \arg\max_{(\mu,\sigma)} \mathbb{E}_{(a_t, a_{t+1}, \ldots, a_{t+H}) \sim \mathcal{N}(\mu, \sigma^2)} \left[ \gamma^H V(z_{t+H}) + \sum_{h=t}^{H-1} \gamma^h R_{\theta^R}(z_h, a_h) \right] \quad (10)$$

where $V(z_{t+H}) = Q_{\theta^Q}(z_{t+H}, \pi_\psi(z_{t+h}))$. We continue this calculation until it reaches the given number of iterations. More details can be found in (Hansen et al., 2022; Williams et al., 2016; 2018).

## 4.2 THEORETICAL ANALYSIS

It is important to measure the quality of the learned representation space. In this section, we show that BS-MPC upper-bounds expected cumulative reward by leveraging value function bounds derived from the on-policy bisimulation metric. This property, absent in TD-MPC, strongly differentiates BS-MPC.

We assume that the learned policy in BS-MPC continuously improves throughout training and eventually converges to the optimal policy $\pi^*$, which supports Theorem 1.

**Theorem 1.** (Theorem 1 in (Zhang et al., 2021)) Let's assume a policy $\pi$ in BS-MPC continuously improves over time, converging to the optimal policy $\pi^*$. Under this assumption, the following bisimulation metric has a least fixed point $\tilde{d}$ and that is a $\pi^*$-bisimulation metric.

$$d(s_i, s_j) = (1 - c)|r_{s_i}^\pi - r_{s_j}^\pi| + c W_p(d)(\mathcal{P}^\pi(\cdot|s_i), \mathcal{P}^\pi(\cdot|s_j)). \quad (11)$$

where $W_p(d)(\mathcal{P}_i, \mathcal{P}_j) = \left( \inf_{\gamma' \in \Gamma(\mathcal{P}_i, \mathcal{P}_j)} \int_{\mathcal{S} \times \mathcal{S}} d(s_i, s_j)^p \, d\gamma'(s_i, s_j) \right)^{1/p}$ and $\Gamma(\mathcal{P}_i, \mathcal{P}_j)$ is the set of all couplings of $\mathcal{P}_i$ and $\mathcal{P}_j$.

Proof can be found in (Zhang et al., 2021). Under this $\pi^*$-bisimulation metric, we can divide the latent space into $n$ partitions based on some $\epsilon > 0$, where $\frac{1}{n} < (1 - c)\epsilon$. Let $\phi$ represent the encoder that maps each original state from the state space $\mathcal{S}$ to a corresponding $\epsilon$-cluster. With these notations, (Zhang et al., 2021) shows the following value bound based on bisimulation metrics.

**Theorem 2.** (Theorem 2 in (Zhang et al., 2021)) Consider an MDP $\bar{\mathcal{M}}$, which is formed by clustering states within an $\epsilon$-neighborhood, along with an encoder $\phi$ that maps states from the original MDP $\mathcal{M}$ to these clusters. Under the same assumption in Theorem 1, optimal value functions for the two MDPs are bounded by

$$|V^*(s) - V^*(\phi(s))| \leq \frac{2\epsilon + 2\mathcal{L}}{(1 - \gamma)(1 - c)} \quad (12)$$

where $\mathcal{L} := \sup_{s_i, s_j \in \mathcal{S}} |\, \|\phi(s_i) - \phi(s_j)\| - \tilde{d}(s_i, s_j)|$ is the learning error for encoder $\phi$. Note that this theorem assumes access to the true dynamics model $\mathcal{P}$ and reward function $\mathcal{R}$.

Proof can also be found in (Zhang et al., 2021). This result demonstrates that the optimal value function in the original state space and the optimal value function in the latent space, projected by the $\pi^*$-bisimulation metric, is bounded from above. Leveraging Theorem 1 and Theorem 2, we can bound the cumulative reward of a trajectory under the original MDP $\mathcal{M}$ and the latent MDP $\bar{\mathcal{M}}$.

**Theorem 3.** Consider a trajectory $\tau = (s_0, a_0, s_1, a_1, \ldots, s_{H-1}, a_{H-1}, s_H)$ in the original state space $\mathcal{S}$, and its corresponding encoded trajectory $\phi(\tau) = (z_0, a_0, z_1, a_1, \ldots, z_{H-1}, a_{H-1}, z_H)$, where $a_k \sim \pi^*(\cdot \mid s_k)$ and $z_k = \phi(s_k)$, with $\phi$ defined as in Theorem 2. Under the same assumption as in Theorem 1 and Theorem 2, the following expected cumulative rewards

$$\mathbf{S}(\tau) = \mathbb{E}_\tau \left[ \gamma^H V^*(s_H) + \sum_{h=0}^{H-1} \gamma^h R(s_h, a_h) \right],$$

$$\mathbf{S}(\phi(\tau)) = \mathbb{E}_\tau \left[ \gamma^H V^*(\phi(s_H)) + \sum_{h=0}^{H-1} \gamma^h R(\phi(s_h), a_h) \right]$$

can be bounded as follows:

$$|\mathbf{S}(\tau) - \mathbf{S}(\phi(\tau))| \leq \frac{2\gamma^H(\epsilon + \mathcal{L})}{(1-\gamma)(1-c)} + \frac{2\epsilon(1-\gamma^H)}{(1-\gamma)(1-c)}. \tag{13}$$

Proof can be found in Appendix A.1. This theorem states that if the cluster radius $\epsilon$ and the encoder error $\mathcal{L}$ are sufficiently small, the learned representation space $\mathcal{Z}$ does not change the original cumulative rewards over the same trajectory $\tau$. This suggests that the latent space retains essential information from the original space.

## 4.3 DIFFERENCE BETWEEN BS-MPC AND TD-MPC

The main difference between BS-MPC and TD-MPC lies in its objective function and computation flow. Specifically, BS-MPC updates the encoder parameter by minimizing MSE loss between on-policy bisimulation metric and $\ell_1$-distance in the latent space at every time step $k$, which is shown in Eq. 7. These differences result in the following improvements.

**Improved training stability** In TD-MPC, the encoder is only updated indirectly through gradients propagated from the latent dynamics loss, as the objective function lacks explicit encoder loss term and only consists of the first three terms in Eq. 8. This indirect update makes it challenging to effectively optimize the encoder parameters at each training step, potentially leading to significant inconsistencies in the latent dynamics and destabilizing the learning process. Such model inconsistencies have also been reported in Zhao et al. (2023). In contrast, BS-MPC has an explicit encoder loss in its objective function thanks to the inclusion of bisimulation metric loss. This allows the gradients of the encoder parameters to be directly computed from the objective function, ensuring continuous improvement of the encoder. As a result, the learned encoder effectively maps the original state $s$ to the latent space $z$, leading to reduced model discrepancies. The encoder update difference is shown in Fig. 2a and Fig. 2b.

**Theoretical support of the latent space** The encoder in BS-MPC generates a latent representation $\mathcal{Z}$ where the $\ell_1$-distance corresponds to the bisimulation metric. This indicates that the encoder efficiently filters out irrelevant information from the original state $s$ and preserves intrinsic details in the latent state $z$. Consequently, BS-MPC exhibits robust resilience to noise. Additionally, this property guarantees that the cumulative reward difference between the learned representation space and the original space over a trajectory is upper-bounded, as discussed in Section 4.2. In contrast, the encoder in TD-MPC lacks theoretical guarantees in its learned representation space, potentially leading to the projection of irrelevant details into the latent space. This absence of theoretical validity in the encoder contributes to increased sensitivity to noise, as shown in our experimental results (Section 5.3).

**Ease of parallelization** TD-MPC predicts the latent state $\hat{z}_{k+1}$ by applying the dynamics model to the previous predicted latent state $\hat{z}_k$, introducing a sequential dependency that hinders parallel computation (see Lines 12 to 17 of Algorithm 2 in Hansen et al. (2022)). In contrast, BS-MPC generates the predicted latent state $\hat{z}_{k+1}$ by encoding the current state into the latent state $z_k = h_{\theta^h}(s_k)$ and using it as input to the dynamics model, which removes the sequential dependency and allows for parallel computation across time steps. Fig. 2a and Fig. 2b show the calculation flow difference between TD-MPC and BS-MPC. Consequently, BS-MPC achieves faster computational times compared to TD-MPC.

## 5 EXPERIMENTS

We evaluate BS-MPC across various continuous control tasks using the DeepMind Control Suite (DM Control (Tassa et al., 2018)). The inputs in these experiments include both high-dimensional state vectors and images, with some tasks set in sparse-reward environments. The objective of this section is to demonstrate that BS-MPC maintains its performance over time and remains robust to noise. Additionally, we aim to confirm that it outperforms TD-MPC in terms of computational efficiency. In this experiment, we focus specifically on comparing BS-MPC with TD-MPC. **For a fair comparison, BS-MPC uses the same model architecture as TD-MPC, with an identical**

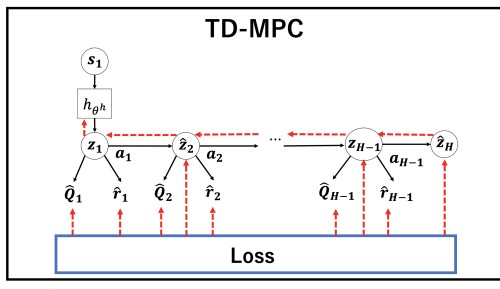 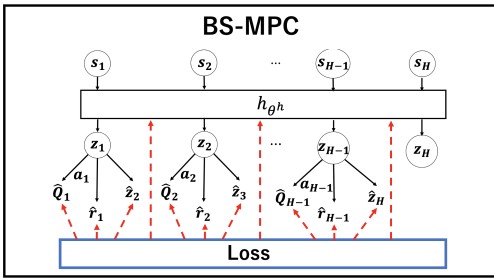

**(a)** TD-MPC calculation flow          **(b)** BS-MPC calculation flow

**Figure 2:** Calculation flow comparison. The black line shows the forward calculation flow, and the red arrows represent the gradient of $\theta^h$. While TD-MPC needs sequential calculation in its forward computational flow, BS-MPC can process all the calculation parallel. Moreover, BS-MPC has explicit encoder loss in its cost function, so its derivative directly updates the parameters of the encoder. Note that TD-MPC only encodes the original observation at the initial time step and predicts latent states by using the latent dynamics model.

**number of parameters and tuning parameters set to the same values. We also run BS-MPC and TD-MPC using the same random seeds.** The only difference between BS-MPC and TD-MPC is explicit encoder loss in the objective function with an additional parameter $c_4$. We adopt the same environmental settings as those used in the original TD-MPC paper (Hansen et al., 2022). Detailed experimental configurations are provided in Appendix D.

For the baselines, we use the model-free RL algorithm SAC (Haarnoja et al., 2018a;b; 2019), the model-based RL algorithm Dreamer-v3 (Hafner et al., 2024), and the planning-based model-based RL approach TD-MPC Hansen et al. (2022) and its successor TD-MPC2 Hansen et al. (2024), evaluating them on both state-based and image-based tasks. Note that we use a model with 5 million parameters for TD-MPC2 because it is their default model. In addition to these algorithms, we also compare BS-MPC with DrQ-v2 (Yarats et al., 2022) and CURL (Laskin et al., 2020) on image-based tasks. We publicly release the value of episode return at each time step and code for training BS-MPC agents.

## 5.1 RESULT ON STATE-BASED TASKS

We evaluate BS-MPC across 26 diverse continuous control tasks with state inputs, comparing its performance to other baseline methods. In this setting, agents have direct access to all internal states of the environment.

Fig. 3 shows the average performance of each algorithm across 10 tasks, along with the individual scores from 9 specific tasks. We ran 10 million time steps for the dog experiment and 8 million time steps for the humanoid experiment. For the other tasks, the experiments were run for 4 million time steps. The results demonstrate that BS-MPC consistently outperforms existing model-based and model-free reinforcement learning methods, particularly in high-dimensional environments. On complex tasks involving dog and humanoid environments, BS-MPC significantly outperforms TD-MPC, SAC, and Dreamer-v3. In particular, BS-MPC achieves higher episode returns early in training and maintains superior performance throughout, whereas the other methods either plateau or display instability. TD-MPC performs well in the early stages of training, achieving competitive results up to approximately 1 million steps. However, in many tasks, its performance suddenly collapses after this point, leading to high variance and reduced episode returns. Both BS-MPC and TD-MPC2 resolve the issue of performance divergence observed in TD-MPC; however, TD-MPC2 requires many more parameters and employs more complex model architectures. Additionally, TD-MPC2 requires significantly more computation time than both TD-MPC and BS-MPC due to its reliance on discrete regression for optimizing the reward and value function models. It is important to note that BS-MPC and TD-MPC share the exact same model architecture, hyperparameters, and number of parameters. The only difference between them lies in the cost function: BS-MPC explicitly minimizes the bisimulation metric loss at every time step to train the encoder, whereas TD-MPC only calculates the encoder loss at the initial time step. Appendix. B shows all the results from 26

continuous control tasks, computation time comparison, and detailed analysis of the training failure of TD-MPC.

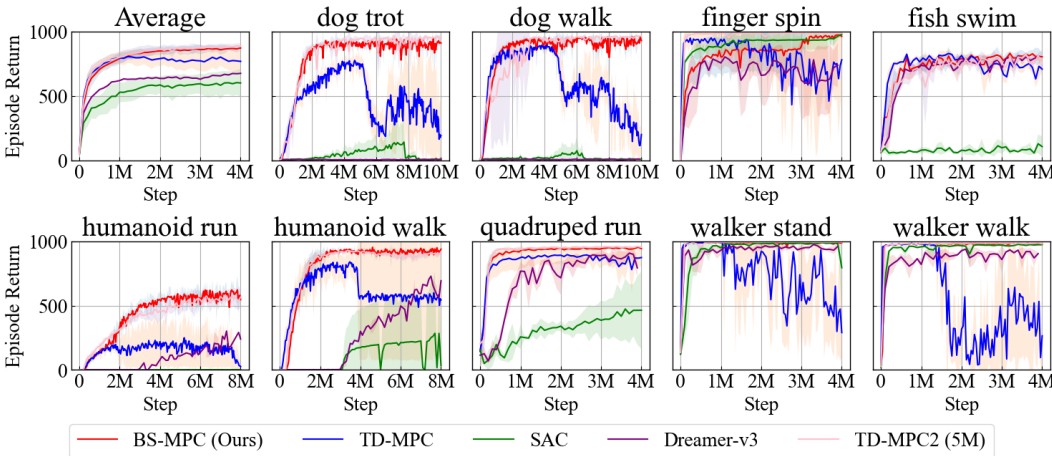

**Figure 3:** Performance comparison on the average over 26 state-based tasks and 9 DM Control tasks with state input. At each evaluation step, the episode return is computed over 10 episodes. The results are averaged over 3 seeds, with shaded regions representing the standard deviation. Results for SAC and Dreamer-v3 are obtained from (Hansen et al., 2024), and results for TD-MPC are reproduced using their official code with the same architecture and hyperparameters for BS-MPC. We use the same seeds for evaluation.

## 5.2 RESULTS ON IMAGE-BASED TASKS

Next, we evaluate BS-MPC and other baseline methods on image-based tasks from 10 DM Control environments. In these tasks, the encoders of both BS-MPC and TD-MPC are modeled by CNN to project high-dimensional image data into a compact latent space. To ensure a fair comparison, we use the same model architecture, hyperparameters, and number of parameters for both BS-MPC and TD-MPC, and we use identical seeds for evaluation. We run 3 million environmental steps for all tasks. Fig. 4 shows the results across 10 image-based tasks. BS-MPC demonstrates performance competitive with TD-MPC, DrQ-v2 and Dreamer-v3, consistently outperforming CURL and SAC. TD-MPC2 converges faster than BS-MPC in certain tasks (e.g., quadruped-run and walker-run); however, BS-MPC achieves comparable or slightly better performance overall with fewer parameters than TD-MPC2.

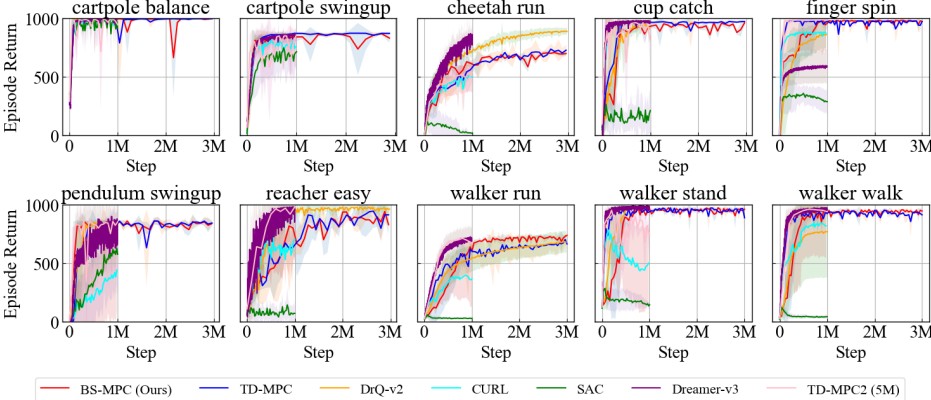

**Figure 4:** Performance comparison on 10 DM Control image-based tasks. At each evaluation step, the episode return is computed over 10 episodes. The results are averaged over 3 seeds, with shaded regions representing the standard deviation. Results for DrQ-v2 are obtained from their official results, and results for CURL, SAC and Dreamer-v3 are obtained from Dreamer-v3 code (Hafner et al., 2024).

## 5.3 RESUTLS ON IMAGE INPUT WITH DISTRACTIONS

Finally, we evaluate the robustness of the proposed method in the presence of distracting information. The goal of this experiment is to show that BS-MPC has better resilience against irrelevant data in the input. We benchmark BS-MPC on 5 DM Control tasks by introducing irrelevant information into the input as noise. Following (Zhang et al., 2018; 2021), driving videos from the Kinetics dataset (Kay et al., 2017) are used as background for the original images. In this experiment, the same parameters and architecture as in Section 5.2 are employed, and the performance of BS-MPC is compared to that of TD-MPC.

Figure 5 shows the experimental results, which reveal that BS-MPC constantly outperforms TD-MPC in every environment. Since TD-MPC does not have an explicit objective function for its encoder, its encoder simply learns representation space to keep the latent dynamics consistent. Therefore, its encoder struggles to filter out the noise during training. BS-MPC, however, learns its encoder by minimizing the bisimulation metric loss to retain bisimulation information in the learned representation space. This architectural modification enhances performance and increases robustness to noise compared to TD-MPC.

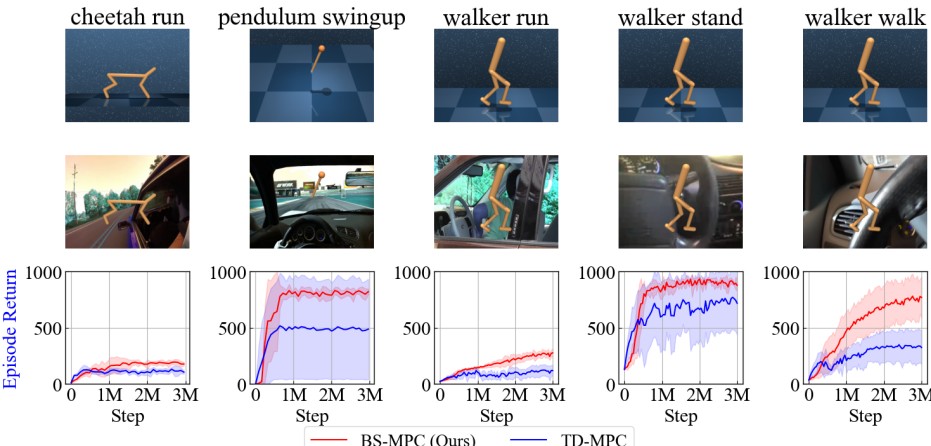

**Figure 5:** Performance comparison on 5 DM Control image-based tasks with distracted information from Kinetics dataset. At each evaluation step, the episode return is computed over 10 episodes. The results are averaged over 5 seeds, with shaded regions representing the standard deviation. (Top) Original Image. (Middle) Distracted Image. (Bottom) Performance results. BS-MPC constantly outperforms TD-MPC when the input is disturbed.

## 6 CONCLUSION

In this paper, we propose a novel model-based reinforcement learning method called Bisimulation Metric for Model Predictive Control (BS-MPC). While inheriting several properties from TD-MPC, our approach differentiates it from the previous method in three key areas: inclusion of explicit encoder loss term, adaptation of bisimulation metric, and parallelizing the computational flow. These improvements stabilize the learning process and enhance the model's robustness to noise while reducing the training time. Experimental results on continuous control tasks from DM Control demonstrate that BS-MPC has superior stability and robustness, whereas TD-MPC and other baselines fail to achieve comparable performance.

**Limitations.**   Despite the theoretical foundations and experimental results supporting BS-MPC, it has one notable limitation: the need for extensive parameter tuning of $c_4$ across different environments. In this paper, we employ a grid search to identify the optimal parameter values; however, future research should focus on developing methods for automatic parameter adjustment.

## 7 ACKNOWLEDGEMENTS

This research was supported by TIER IV, Inc. through the Student Research Scholarship program.

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

## A    PROOF AND ANALYSIS

In this section, we provide proof of our statement and some analysis to give a theoretical difference between BS-MPC and TD-MPC.

### A.1    PROOF OF THEOREM 3

Our proof uses the following Lemma.

**Lemma 1.** Assume action $a$ is sampled from optimal policy $a \sim \pi^*(\cdot|s)$. With the same assumption in Theorem 2, the difference between $R(s, a)$ and $R(\phi(s), a)$ under the $\pi^*$-bisimulation metric has the following upper bound.

$$(1 - c)|R(s, a) - R(\phi(s), a)| \leq 2\epsilon \tag{14}$$

*Proof.* From Theorem 1, the fixed point $\tilde{d}$ satisfies

$$\tilde{d}(s, \phi(s)) = (1 - c)|R(s, a) - R(\phi(s), a)| + cW_p(d)(\mathcal{P}^{\pi^*}(\cdot|s_i), \mathcal{P}^{\pi^*}(\cdot|s_j)). \tag{15}$$

Since the second term is always positive, we can get

$$(1 - c)|R(s, a) - R(\phi(s), a)| \leq \tilde{d}(s, \phi(s))$$
$$\leq 2\epsilon \tag{16}$$

$\square$

**Theorem 3.** Consider a trajectory $\tau = (s_0, a_0, s_1, a_1, \ldots, a_{H-1}, s_H)$ in the original state space $\mathcal{S}$, and its corresponding encoded trajectory $\phi(\tau) = (z_0, a_0, z_1, a_1, \ldots, a_{H-1}, z_H)$, where $a_k \sim \pi^*(\cdot|s_k)$ and $z_k = \phi(s_k)$, with $\phi$ defined in Theorem 2. Under the assumption that both the reward model and dynamics model have no approximation error, the cumulative rewards $\mathbf{S}(\tau) = \mathbb{E}_\tau \left[ \gamma^H V^*(s_H) + \sum_{h=0}^{H-1} \gamma^h R(s_h, a_h) \right]$ and $\mathbf{S}(\phi(\tau)) = \mathbb{E}_\tau \left[ \gamma^H V^*(\phi(s_H)) + \sum_{h=0}^{H-1} \gamma^h R(\phi(s_h), a_h) \right]$ can be bounded as follows.

$$|\mathbf{S}(\tau) - \mathbf{S}(\phi(\tau))| \leq \frac{2\gamma^H(\epsilon + \mathcal{L})}{(1 - \gamma)(1 - c)} + \frac{2\epsilon(1 - \gamma^{H-1})}{(1 - \gamma)(1 - c)} \tag{17}$$

*Proof.* Simply calculating the difference between $\mathbf{S}(\tau)$ and $\mathbf{S}(\phi(\tau))$

$$|\mathbf{S}(\tau) - \mathbf{S}(\phi(\tau))| = \left| \mathbb{E}_\tau \left[ \gamma^H \left( V^*(s_H) - V^*(\phi(s_H)) \right) + \sum_{h=0}^{H-1} \gamma^h (R(s_h, a_h) - R(\phi(s_h), a_h)) \right] \right|$$

$$= \left| \mathbb{E}_\tau \left[ \gamma^H \left( V^*(s_H) - V^*(\phi(s_H)) \right) \right] + \mathbb{E}_\tau \left[ \sum_{h=0}^{H-1} \gamma^h (R(s_h, a_h) - R(\phi(s_h), a_h)) \right] \right|$$

$$\leq \left| \mathbb{E}_\tau \left[ \gamma^H \left( V^*(s_H) - V^*(\phi(s_H)) \right) \right] \right| + \left| \mathbb{E}_\tau \left[ \sum_{h=0}^{H-1} \gamma^h (R(s_h, a_h) - R(\phi(s_h), a_h)) \right] \right|$$
$$\text{(Triangle inequality)}$$

$$\leq \mathbb{E}_\tau \left[ \gamma^H |V^*(s_H) - V^*(\phi(s_H))| \right] + \mathbb{E}_\tau \left[ \sum_{h=0}^{H-1} \gamma^h |R(s_h, a_h) - R(\phi(s_h), a_h)| \right]$$
$$\text{(Jensen's inequality)}$$

$$= \gamma^H \mathbb{E}_\tau \left[ |V^*(s_H) - V^*(\phi(s_H))| \right] + \sum_{h=0}^{H-1} \gamma^h \mathbb{E}_\tau \left[ |R(s_h, a_h) - R(\phi(s_h), a_h)| \right]$$

$$\leq \frac{2\gamma^H(\epsilon + \mathcal{L})}{(1 - \gamma)(1 - c)} + \frac{2\epsilon}{1 - c} \sum_{h=0}^{H-1} \gamma^h \quad \text{(From Theorem 2 and Lemma 1)}$$

$$\leq \frac{2\gamma^H(\epsilon + \mathcal{L})}{(1 - \gamma)(1 - c)} + \frac{2\epsilon(1 - \gamma^{H-1})}{(1 - \gamma)(1 - c)}$$
$$\tag{18}$$

□

# B ADDITIONAL EXPERIMENTAL RESULTS

## B.1 ALL STATE-BASED TASKS RESULT

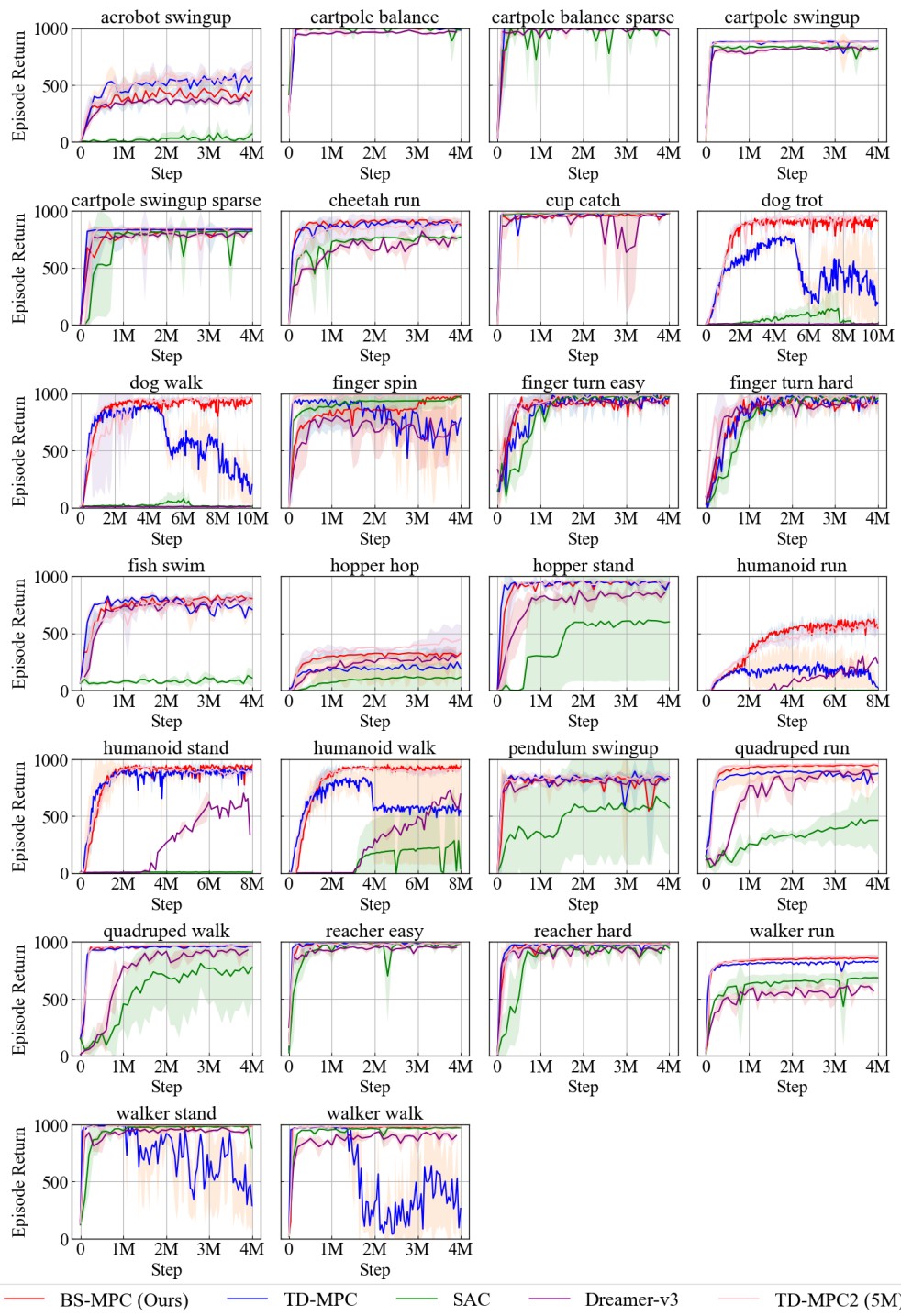

**Figure 6:** State-based tasks result from DMControl Suite. Performance comparison on 26 DM Control tasks with state input. At each evaluation step, the episode return is computed over 10 episodes. The results are averaged over 3 seeds, with shaded regions representing the standard deviation.

## B.2 COMPUTATIONAL TIME

Table. 1 shows the computational time of BS-MPC and TD-MPC. We use RTX-4090 for our experiments.

**Table 1:** Computational time between BS-MPC and TD-MPC in state-based tasks. The table shows how many hours the training takes.

|         | Cartpole-Swingup | Cheetah-run | Finger-Spin | Walker-run |
|---------|------------------|-------------|-------------|------------|
| BS-MPC  | **2.0**          | **4.0**     | **8.3**     | **8.3**    |
| TD-MPC  | 2.4              | 4.8         | 10.0        | 10.1       |

## B.3 DETAILED ANALYSIS FOR THE FAILURE OF TD-MPC

In this section, we analyze the reason why TD-MPC failed to achieve the same performance as BS-MPC in our experiments. First, we look at the losses of both BS-MPC and TD-MPC in the Humanoid-Walk environment. Fig 7 shows the average value of each loss over the batches. From this image, it can be observed that the consistency loss in TD-MPC gradually increases and diverges. In contrast, BS-MPC has much smaller values for every component, resulting in more stable performance. One possible cause is that TD-MPC fails to learn the encoder, leading to large errors, which in turn result in the failure to train the latent dynamics model properly. BS-MPC, on the contrary, has explicit encoder loss in its objective function, thus enabling it to actively update the encoder. Fig 8 shows the learned Q values and gradient norm of the objective function. As it shows, TD-MPC is vulnerable to exploding gradients, which leads to the divergence of the loss. Moreover, the learned Q values drop significantly when the gradient norm explodes.

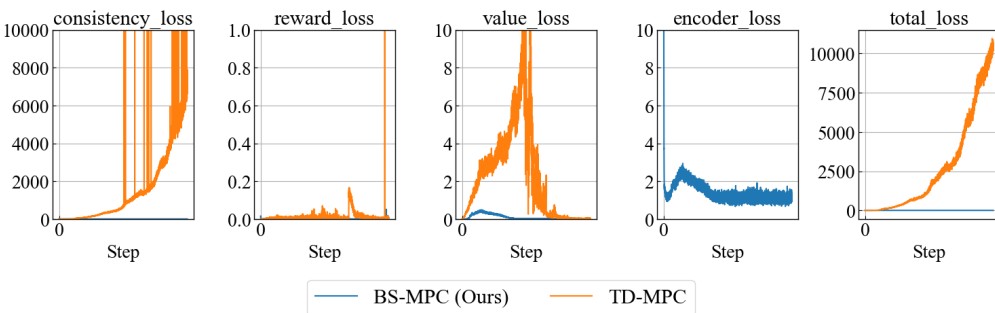

**Figure 7:** Comparative analysis of loss components between BS-MPC and TD-MPC across training steps in Humanoid-walk environment. Each graph presents different loss types—consistency loss, reward loss, value loss, encoder loss, and total loss—plotted against training steps. Note that TD-MPC does not have encoder loss, and it only exists in BS-MPC. See Eq. 8 for more details.

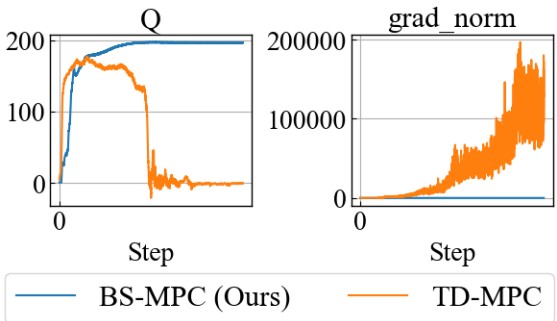

**Figure 8:** Average values of the learned Q functions and gradient norm of the loss function between BS-MPC and TD-MPC across training steps in Humanoid-walk environment.

## C  BS-MPC TRAINING ALGORITHM FLOW

In this section, we describe the algorithm flow of BS-MPC and compare it with TD-MPC.

### C.1  ALGORITHM FLOW

The training algorithm flow is described in Algorithm. 1.

---

**Algorithm 1** BS-MPC (Model training)

---

**Require:** $\theta = [\theta^h, \theta^R, \theta^Q, \theta^d], \psi$: randomly initialized network parameters,
Episode Length $L$, Number of parameter update $K$, Buffer $\mathcal{B}$

1: **while** the training is not complete **do**
2:   *// Collect episode*
3:   **for** $t = 0 \dots L$ **do**
4:     $a_t \sim BSMPC(h_{\theta^h}(s_t))$ {Compute action with BS-MPC}
5:     $(s_{t+1}, r_t) \sim \mathcal{P}(s_t, a_t), \mathcal{R}(s_t, a_t)$ {Execute action against the environment}
6:     $\mathcal{B} \leftarrow \mathcal{B} \cup (s_t, a_t, r_t, s_{t+1})$ {Add to buffer}
7:   **end for**
8:   *// Update model parameters*
9:   $\theta_0 = \theta$ {Initialize $\theta_0$ with current parameter}
10:   **for** $k = 0 \dots K$ **do**
11:     $\{s_{t:t+H+1}, a_{t:t+H}, r_{t:t+H}\} \sim \mathcal{B}$ {Sample a trajectory with horizon $H$ from the buffer $\mathcal{B}$}
12:     $z_{t:t+H+1} = h_{\theta_k^h}(s_{t:t+H+1})$ {Encode all observations with online encoder}
13:     $\hat{r}_{t:t+H} = R_{\theta_k^R}(z_{t:t+H}, a_{t:t+H})$ {Estimated rewards}
14:     $\hat{Q}_{t:t+H} = Q_{\theta_k^Q}(z_{t:t+H}, a_{t:t+H})$ {Estimated state-action value}
15:     $\hat{z}_{t+1:t+H+1} = d_{\theta_k^d}(z_{t:t+H}, a_{t:t+H})$ {Estimated next latent state}
16:     $\theta_{k+1} = \arg\min_{\theta_k} \mathcal{L}(\theta_k)$ {Update $\theta_k$ by minimizing Eq. 7}
17:     $\psi_{k+1} = \arg\min_{\psi_k} \mathcal{J}_\pi(\psi_k)$ {Update $\psi_k$ by minimizing Eq. 9}
18:   **end for**
19:   $\theta = \theta_{K+1}$ {Update current parameter}
20: **end while**

---

### C.2  COMPARISON WITH TD-MPC

As discussed in Section 4.3, BS-MPC facilitates parallel computation. Algorithm 2 outlines the calculation flow of TD-MPC. While TD-MPC shares many similarities with BS-MPC, the primary distinction lies in how it computes the estimated values and the model cost $\mathcal{L}$.

In BS-MPC, the observed state variables $s_{t:t+H}$ over $H$ steps are first projected into a sequence of latent states $z_{t:t+H}$. Subsequently, the rewards, state-action values, and predicted next states for these $H$ latent states are computed collectively. Since all calculations are performed simultaneously across the $H$ steps, parallel computation is effectively utilized, resulting in high computational efficiency. This computation process is detailed in Lines 12 to 15 of Algorithm 1.

Conversely, in TD-MPC, only the initial state $s_t$ is encoded into the latent state $z_t$ (see Line 12 of Algorithm 2), and the subsequent latent states $\hat{z}_{t+1}$ are computed sequentially using the latent dynamics. As a result, TD-MPC requires sequential computation when calculating rewards, state-action values, and the cost $\mathcal{L}$, which limits its ability to leverage parallel computation. This sequential computation process is described in Lines 14 to 19 of Algorithm 2.

In summary, BS-MPC obtains the sequence of latent states $z_{t:t+H}$ by encoding the entire sequence of observed states $s_{t:t+H}$ using the encoder $h_{\theta^h}$. In contrast, TD-MPC encodes only the initial state $s_t$ into $z_t$ and derives the remaining latent states $z_{t+1:t+H}$ sequentially using the latent dynamics

$d_{\theta_k^d}$. Therefore, while BS-MPC enjoys parallel computations to speed up its calculation, TD-MPC suffers from the bottleneck of the sequential computation in the cost calculation.

---

**Algorithm 2** TD-MPC (Model training)

---

**Require:** $\theta = [\theta^h, \theta^R, \theta^Q, \theta^d], \psi$: randomly initialized network parameters,
  Episode Length $L$, Number of parameter update $K$, Buffer $\mathcal{B}$

1: **while** the training is not complete **do**
2:    *// Collect episode*
3:    **for** $t = 0 \ldots L$ **do**
4:       $a_t \sim TDMPC(h_{\theta^h}(s_t))$ {Compute action with TD-MPC}
5:       $(s_{t+1}, r_t) \sim \mathcal{P}(s_t, a_t), \mathcal{R}(s_t, a_t)$ {Execute action against the environment}
6:       $\mathcal{B} \leftarrow \mathcal{B} \cup (s_t, a_t, r_t, s_{t+1})$ {Add to buffer}
7:    **end for**
8:    *// Update model parameters*
9:    $\theta_0 = \theta$ {Initialize $\theta_0$ with current parameter}
10:   **for** $k = 0 \ldots K$ **do**
11:      $\{s_{t:t+H+1}, a_{t:t+H}, r_{t:t+H}\} \sim \mathcal{B}$ {Sample a trajectory with horizon $H$ from the buffer $\mathcal{B}$}
12:      $\hat{z}_t = h_{\theta_k^h}(s_t)$ {Encode the initial observation with online encoder}
13:      $\mathcal{L} = 0$ {Initialize the cost}
14:      **for** $i = t \ldots t + H$ do **do**
15:         $\hat{r}_i = R_{\theta_k^R}(z_i, a_i)$ {Estimated rewards}
16:         $\hat{q}_i = Q_{\theta_k^Q(z_i, a_i)}$ {Estimated state-action value}
17:         $\hat{z}_{i+1} = d_{\theta_k^d}(\hat{z}_i, a_i)$ {Estimated next latent state}
18:         $\mathcal{L} \leftarrow \mathcal{L} + \lambda^{i-t} \mathcal{L}_i(\hat{z}_{i+1}, \hat{r}_i, \hat{q}_i, a_i)$ {Add to the cost}
19:      **end for**
20:      $\theta_{k+1} = \arg\min_{\theta_k} \mathcal{L}(\theta_k)$ {Update $\theta_k$ by minimizing Eq. 7}
21:      $\psi_{k+1} = \arg\min_{\psi_k} \mathcal{J}_\pi(\psi_k)$ {Update $\psi_k$ by minimizing Eq. 9}
22:   **end for**
23:   $\theta = \theta_{K+1}$ {Update current parameter}
24: **end while**

---

## D   IMPLEMENTATION DETAILS

Here we give details about the hyper-parameters and model architectures.

### D.1   HYPERPARAMETERS

**Shared Parameters**: First, we outline the parameters that are common to both TD-MPC and BS-MPC. They are described in Table. 2.

**BS-MPC specific parameters**: Next, we list the parameter that is used for tuning the weight for bisimulation metric loss $(c_4)$. We change the value based on the environment and tune the weighting coefficient $c_4$ across $10^{-8}, 0.0001, 0.001, 0.01, 0.1, 0.5$ with grid search. All of the numbers are listed in Table. 3.

### D.2   MODEL ARCHITECTURE

In our experiments, both BS-MPC and TD-MPC utilize the same model architecture and the number of trainable parameters. We employ multi-layer perceptrons (MLPs) to represent the underlying environment models $\mathcal{P}$ and $\mathcal{R}$, the state-action value function $Q$, and the policy $\pi$. The architecture details are shown in Table. 4. More details can be found in our official code.

**Table 2:** Hyperparameters used for TD-MPC and BS-MPC in the experiment.

| Hyperparameter | Value |
|---|---|
| Discount factor ($\gamma$) | 0.99 |
| Seed steps | 5,000 |
| Replay buffer size | 1,000,000 (state-based tasks) |
|  | 100,000 (image-based tasks) |
| Sampling technique | Uniform Sampling |
| Planning horizon ($H$) | 5 |
| Initial parameters ($\mu^0, \sigma^0$) | (0, 2) |
| Population size | 512 |
| Elite fraction | 64 |
| MPPI Update Iterations | 12 (Humanoid, Dog) |
|  | 6 (otherwise) |
| Policy fraction | 5% |
| Number of particles | 1 |
| Temperature ($\tau$) | 0.5 |
| Latent dimension | 100 (Humanoid, Dog) |
|  | 50 (otherwise) |
| Learning rate | 3e-4 (pixels) |
|  | 1e-3 (otherwise) |
| Optimizer ($\theta$) | Adam ($\beta_1 = 0.9, \beta_2 = 0.999$) |
| Temporal coefficient ($\lambda$) | 0.5 |
| Reward loss coefficient ($c_1$) | 0.5 |
| Value loss coefficient ($c_2$) | 0.1 |
| Consistency loss coefficient ($c_3$) | 0.5 |
| Exploration schedule ($\epsilon$) | $0.5 \rightarrow 0.05$ (25k steps) |
| Planning horizon schedule | $1 \rightarrow 5$ (25k steps) |
| Batch size | 512 (State-based tasks) |
|  | 256 (Image-based tasks) |
| Momentum coefficient ($\zeta$) | 0.99 |
| Steps per gradient update | 1 |
| Target parameter $\bar{\theta}$ update frequency | 2 |

**Table 3:** Bisimulation metric parameter used in the experiment.

| Environment | Value ($c_4$) |
|---|---|
| Acrobot | 0.0001 |
| Cartpole | 0.5 |
| Cheetah | 0.001 |
| Cup | 0.5 |
| Finger | 0.001 |
| Fish | 0.001 |
| Hopper | 0.1 |
| Humanoid | 0.001 |
| Pendulum | 0.01 |
| Quadruped | 0.1 |
| Reacher | 0.01 |
| Walker | 0.001 |
| Dog | $10^{-8}$ |

**Table 4:** Model Architecture used in the experiment.

| Models | Number of Layers | Hidden Dim | Activation |
|---|---|---|---|
| Latent Model Dynamics $\mathcal{P}$ | 3 | 512 | ELU |
| Reward Model $\mathcal{R}$ | 3 | 512 | ELU |
| State-action value function $Q$ | 3 | 512 | ELU + LayerNorm |
| Policy $\pi$ | 3 | 512 | ELU |

