# OpenReview forum: "Bisimulation Metric for Model Predictive Control"
_ICLR.cc/2025/Conference — ICLR 2025 Poster_

### Official Review · Reviewer_seSt · 2024-10-29

**Soundness:** 3
**Presentation:** 3
**Contribution:** 2
**Rating:** 5
**Confidence:** 4

**Summary:**

This paper is concerned with a new, model-based reinforcement learning method, which utilizes bi-simulation metric. This formulation helps with the stability of training and helps with the robustness of the controller. General idea is that they seek to find states that "behave" similarly, and intuition behind it is that one can use similar control input for similar states, which simplifies the controller and make it more interpretable. Authors learn an encoder which maps states of the environment to another domain, in which similar states are identified, and are mapped to the same representation (roughly speaking). Then, this representation is utilized to train a controller. Novelty of this work is to add the encoder loss directly into the training procedure.

**Strengths:**

Paper is well written, all ideas are clearly explained. Overall, paper is mathematically rigorous. Authors do a good job in walking the reader through the preliminaries, highlighting distinctions, and presenting their work.

Furthermore, paper presents more than 20 case studies, which helps immensely in comparing their performance to the state of the art methods.

**Weaknesses:**

Improvements and contributions seem incremental, and overall not that beneficial according to the case studies (Figure 6), one only sees improvements in few case studies (such as humanoid walk, dog walk and trot), and basically identical to TD-MPC in others (such as humanoid stand, pendulum, cheetah).
The only major contribution is adding the bi-simulation metric loss to the loss function, the other two contribution naturally follow from this addition.

As authors have mentioned, the hyper parameters play a huge role, and one wonders how much time is needed to tune these parameters.

**Questions:**

1- Based on your case studies, your method does not seem to change the episode return that much, except for a few cases like dog walk, dog trot and humanoid walk. In your Appendix, you provide a rough explanation of why that may be. Looking at Figure 7 and 8, it appears that loss, and consequently the gradient, explode (in TD-MPC), however, in RL, gradient clipping is used to tackle this issue. When you compared your method to TD-MPC, did you employ gradient clipping for it or not? It does not appear to be a fair comparison if you didn't, and perhaps that is why your method did not do significantly better in other case studies; as loss did not *explode*.

2- I suggest you revise the experiments' section, and run all case studies on TD-MPC2 rather than TD-MPC. I realize it is touched upon in appendix D, however since TD-MPC2 is the updated version, I suspect it would make for a more fair comparison. Moreover, adding a thorough comparison would certainly present your method better; between training time, sample complexity, number of parameters used, and hyper parameter tuning and different configurations; it will strengthen your case if you could show how it might fail. I would also like to know the rationale behind using TD-MPC in the main body and mentioning TD-MPC2 in the appendix.
To the best of my knowledge, TD-MPC2 can have many parameters, since it can be used on different domains. Thus, it is not an apple to apple comparison, unless it is specifically mentioned in the paper.


3- Is there any theoretical results on why your method requires less parameters, and converges faster? or is it mainly based off of experiments? Since Theorem 3 only offers an upper bound for expected cumulative rewards for the optimal policy.

4- I suggest a more rigorous approach for robustness, such as comparing the Lipschitz constant of your controller to TD-MPC's.

5- I believe if you can theoretically confirm in which case studies your method is going to perform better than TD-MPC, it will strengthen your paper significantly.

---

> ### Author Response · Authors · 2024-11-17
> **Author repsonse to reviewer seSt Part1**
>
> We are grateful for the reviewer’s constructive suggestions and provide our responses below.
>
> Q1. Based on your case studies, your method does not seem to change the episode return that much, except for a few cases like dog walk, dog trot and humanoid walk. In your Appendix, you provide a rough explanation of why that may be. Looking at Figure 7 and 8, it appears that loss, and consequently the gradient, explode (in TD-MPC), however, in RL, gradient clipping is used to tackle this issue. When you compared your method to TD-MPC, did you employ gradient clipping for it or not? It does not appear to be a fair comparison if you didn't, and perhaps that is why your method did not do significantly better in other case studies; as loss did not explode.
>
> A1. Yes, we do use gradient clipping for TD-MPC in our experiments. Specifically, the setting file (yaml file) used in our experiments includes a parameter labeled 'clip_grad_norm,' which specifies the gradient clipping threshold. In Figure 8, we present the gradient values before clipping the gradient for transparency.
>
> Although gradient clipping is applied, we observe that the error values continue to grow and eventually explode, leading to degraded performance. This issue highlights the inherent instability in TD-MPC. Moreover, while BS-MPC achieves similar performance to TD-MPC in many cases, it demonstrates significantly superior robustness in noisy environments, as shown in Figure 5.
>
> Finally, we want to emphasize that the primary objective of BS-MPC is to provide a more stable learning process and performance compared to TD-MPC, rather than merely outperforming it across all metrics. This is one of the open problems in TD-MPC as shown in Figure 1.
>
> ---
>
> Q2. I suggest you revise the experiments' section, and run all case studies on TD-MPC2 rather than TD-MPC. I realize it is touched upon in appendix D, however since TD-MPC2 is the updated version, I suspect it would make for a more fair comparison. Moreover, adding a thorough comparison would certainly present your method better; between training time, sample complexity, number of parameters used, and hyper parameter tuning and different configurations; it will strengthen your case if you could show how it might fail. I would also like to know the rationale behind using TD-MPC in the main body and mentioning TD-MPC2 in the appendix.
> To the best of my knowledge, TD-MPC2 can have many parameters, since it can be used on different domains. Thus, it is not an apple to apple comparison, unless it is specifically mentioned in the paper.
>
> A2. Thank you for the suggestion. We agree with the reviewer's idea and are working on this. Since TD-MPC2 does not provide some of the results in image-based tasks, we are currently running their official code.
>
> There are three primary reasons why we chose TD-MPC over TD-MPC2 as our baseline:
>
> - Parameter Count: As the reviewer highlighted, TD-MPC2 requires significantly more parameters compared to both BS-MPC and TD-MPC. While BS-MPC and TD-MPC have approximately 1M parameters, TD-MPC2 requires 5M parameters to achieve comparable performance. This increased complexity substantially raises computational costs, which is a concern given that TD-MPC already incurs high computational overhead.
>
> - Discrete Regression: TD-MPC2 introduces discrete regression to handle variations in reward magnitude and Q-values. While this eliminates the need for certain parameter tuning, it also necessitates discretizing these values, further increasing computational costs. This trade-off made TD-MPC2 less aligned with our goal of balancing performance with computational efficiency.
>
> - Network Complexity: TD-MPC2 employs additional layers and normalization mechanisms, such as SimNorm normalization, which biases the latent state representation towards sparsity and maintains a small ℓ2-norm. While these techniques have shown empirical success, they lack a robust theoretical foundation to justify their inclusion. In contrast, BS-MPC intentionally adopts the same simple network architecture as TD-MPC, augmenting it with a bisimulation metric to regularize the latent space projection. This simplicity aligns with our design philosophy while having preferrable theoretical supports.
>
> ---
>
> Q3. Is there any theoretical results on why your method requires less parameters, and converges faster? or is it mainly based off of experiments? Since Theorem 3 only offers an upper bound for expected cumulative rewards for the optimal policy.
>
> A3. BS-MPC has the exact same number of parameters as TD-MPC as both of them use the exact same architecture. However, since our method has a different computational structure, BS-MPC can enjoy parallel computation when computing and optimizing its cost. This is attributed to the faster computational time. We add Appendix C.2 for more details about the computational structure change.

---

> ### Author Response · Authors · 2024-11-17
> **Author repsonse to reviewer seSt Part2**
>
> Q4- I suggest a more rigorous approach for robustness, such as comparing the Lipschitz constant of your controller to TD-MPC's.
>
> A4. We explored the possibility of computing the Lipschitz constant for both our approach and TD-MPC. However, due to the complexity introduced by the use of neural networks in both methods, deriving a rigorous mathematical analysis proved to be challenging.
>
> To address this limitation, we have added an intuitive explanation in Appendix A.2 to provide insight into why BS-MPC demonstrates improved robustness compared to TD-MPC. This discussion complements the response provided in Q5 and offers a qualitative understanding of the underlying factors.
>
> ---
>
> Q5. I believe if you can theoretically confirm in which case studies your method is going to perform better than TD-MPC, it will strengthen your paper significantly.
>
> A5. While we have not yet established a rigorous theoretical framework to precisely predict such cases, we have included an additional analysis in Appendix A.2. This section explores the factors that contribute to BS-MPC's improved performance in certain scenarios.
>
> We kindly invite the reviewer to review this new addition. In summary, BS-MPC demonstrates superior performance when TD-MPC struggles to train its encoder to effectively capture the essential information in the original state $s$. This provides a partial explanation for BS-MPC's higher scores in noisy environments (Figure 5), where the TD-MPC encoder is more likely to fail to learn an accurate representation.
>
> ---
>
> Q6. As the authors have mentioned, the hyperparameters play a huge role, and one wonders how much time is needed to tune these parameters.
>
> A6. In our experiment, we use the grid search for hyperparameter optimization. Since there are only 6 candidates, it does not take much time to adjust the parameter. We also find smaller values are suitable for more complex environments (e.g. dogs and humanoids) from our experimental results, and this insight helps us to narrow down the range for searching parameters.
>
> ---
>
> Q7. Contribution weakness
>
> A7. We wrote our thoughts in [general reposense](https://openreview.net/forum?id=F07ic7huE3&noteId=x4OIKV1V1Z).

---

> ### Author Response · Authors · 2024-11-20
> **Author repsonse to reviewer seSt Part3**
>
> We have conducted the additional experiments for TD-MPC2 and updated the PDF with the new results. Following the reviewer’s suggestion, we have moved Appendix D to Section 5 to better highlight the experimental results, specifically by including the TD-MPC2 results in the main body of the paper.

---

> > ### Comment · Reviewer_seSt · 2024-11-21
> >
> > I appreciate your response, and your dedication in including new method to compare. However, I believe this work to be too heuristic and incremental, and as I stated, formal theorem on your findings would have strengthen your paper. I will maintain my rating according to the paper's contributions and comments.
> >
> > Minor: I think your new figures are confusing, I suggest to use the same color for each method throughout the paper. TD-MPC2 is pink in Figure 4, yet it is purple in Figure 3.

---

> > > ### Author Response · Authors · 2024-11-21
> > > **Author repsonse to reviewer seSt**
> > >
> > > Thank you for the comment and suggestion. We updated our pdf to change the color of the graph and clarify the diagram. In addition, we continue working on the theory side to see if we can give more theories and statements. We will send the reviewers our new findings as soon as we get new update.

---

> > > > ### Author Response · Authors · 2024-11-26
> > > > **New theorem added**
> > > >
> > > > Upon the comments by reviewer seSt, we added a new theorem that mathematically shows when BS-MPC can outperform TD-MPC. We have updated the paper and uploaded it. The new theorem can be found in Appendix A.2.
> > > >
> > > > Here is a short summary of the theorem.
> > > >
> > > > When we have an original state $s$ and a noisy state $\tilde{s} = s + \xi$, where $xi$ is a noise. Assume $s$ and $\tilde{s}$ are biosimilar. Under some assumptions (specified in the paper), we can get the following upper bound.
> > > >
> > > > - TD-MPC
> > > > $ \|\phi^{\text{TM}}(s) - \phi^{\text{TM}}(\tilde{s})\|_1 \leq K \|\xi\|_1$
> > > >
> > > > - BS-MPC
> > > >  $\|\phi^{\text{BM}}(s) - \phi^{\text{BM}}(\tilde{s})\|_1 \leq \mathcal{L}$
> > > >
> > > > This result suggests that the upper bound of BS-MPC does not depend on the noise $\xi$, whereas the upper bound of TD-MPC depends on the noise $\xi$. This suggests that BS-MPC can identify two biosimilar states, but TD-MPC can classify them as totally different states (The difference between $\phi(s)$ and $\phi(\tilde{s})$ can be very large).
> > > >
> > > > To the best of our knowledge, this is the first work to mathematically show the conditions under which TD-MPC fails and the proposed method succeeds.
> > > > We also highlight that our approach is supported by mathematical foundations that are comparable to or stronger than those of other model-based or TD-MPC family methods [1][2][3][4][5].
> > > >
> > > > [1]. Hansen et al. (2024). TD-MPC2: Scalable, Robust World Models for Continuous Control.
> > > > [2]. Zhao et al. (2023). Simplified Temporal Consistency Reinforcement Learning. In Proceedings of the 40th International Conference on Machine Learning (ICML'23).
> > > > [3]. Yang et al. (2024). MoVie: Visual Model-Based Policy Adaptation for View Generalization. In Proceedings of the 37th International Conference on Neural Information Processing Systems (NeurIPS '23).
> > > > [4]. Zheng et al. (2024). TACO: Temporal Latent Action-Driven Contrastive Loss for Visual Reinforcement Learning. In Proceedings of the 37th International Conference on Neural Information Processing Systems (NeurIPS '23).
> > > > [5]. Ji et al. (2023). Dual Policy-Based TD-Learning for Model Predictive Control. In Proceedings of the 2023 International Conference on Artificial Intelligence in Information and Communication (ICAIIC),

---

> ### Comment · Reviewer_seSt · 2024-11-26
>
> Thank you for adding this theorem.
>
> I believe this theorem was added in haste, I originally asked the authors that if you are claiming robustness, you should do Lipschitz constant analysis. To which authors responded that it is intractable/computationally expensive, without knowing the Lipschitz constants, how can you claim you are more robust?
>
> Moreover, equation 20 is misleading, Lipschitz continuity also applies to your encoding, and you also suffer from noise, if you employ global Lipschitz continuity. I personally think this theorem would hurt your paper rather than helping it.

---

### Official Review · Reviewer_hjrQ · 2024-10-31

**Soundness:** 3
**Presentation:** 3
**Contribution:** 2
**Rating:** 6
**Confidence:** 4

**Summary:**

This paper considers model based reinforcement learning and proposes bisimulation metric to improve over temporal differential MPC method. The authors show theoretical analysis of the expected cumulative rewards in the latent space, and empirically demonstrate enhancement over TD-MPC and other baselines on several continuous control tasks.

**Strengths:**

The paper is clearly written and well presented. The proposed bisimulation metric seems to work well on the experiments considered, compared to TD-MPC and other baselines. The supplementary sections are comprehensive.

**Weaknesses:**

The novelty of the paper seems ambiguous. It seems that both on-policy bisimulation and TD-MPC methods are well studied for model based RL, and the authors plug bisimulation into TD-MPC.

There are several typos in the paper.
“In BS-MPC, the latent dynamics are modeled using an MLP. We also model the latent dynamics model with an MLP” I believe BS-MPC should be TD-MPC.
“we sample M action sets from Gaussian distribution N (μ0, σ0) based on the initial meanμ0 and standard deviation σ0” Missing spacing between mean and \mu_0

**Questions:**

“We assume that the learned policy in BS-MPC continuously improves throughout training and eventually converges to the optimal policy π∗, which supports Theorem 1.”
This seems to be a very strong assumption. For example, by looking at the training curve, the return does not improve monotonically, and we have no information about if the learned policy is converging to the optimal policy. How do you explain such a strong assumption? Is it possible to remove it for the theoretical results?

In Fig. 4, why do all non-MPC based methods only have results till 10M steps?

---

> ### Author Response · Authors · 2024-11-17
> **Author Response to reviewer hjrQ**
>
> We are grateful for the reviewer’s constructive suggestions and provide our responses below.
>
> Q1: “We assume that the learned policy in BS-MPC continuously improves throughout training and eventually converges to the optimal policy π∗, which supports Theorem 1.” This seems to be a very strong assumption. For example, by looking at the training curve, the return does not improve monotonically, and we have no information about if the learned policy is converging to the optimal policy. How do you explain such a strong assumption? Is it possible to remove it for the theoretical results?
>
> A1: Yes! We believe we can loosen this assumption by adapting the theory from the Robust Bisimulation Metric[1]. In this paper, they enable the bisimulation metric to have the same guarantee with some more realistic assumptions. In our paper, we choose the optimal policy assumption \pi^* because we would like to make the algorithm and proof simpler and straightforward. However, as the reviewer suggested, we believe our method can be more adaptable with realistic assumptions by using their robust bisimulation metric.
>
> [1]. Kemertas, M., & Aumentado-Armstrong, T. (2021). "Towards Robust Bisimulation Metric Learning." https://arxiv.org/pdf/2006.10742
>
> ---
>
> Q2: In Fig. 4, why do all non-MPC based methods only have results till 10M steps?
>
> A2: In image-based experiments (fig 4), we cite our baseline results from their official papers. I assume the computational cost is very expensive and authors just run their code until 1M. We run BS-MPC and TD-MPC until 3M to show the stability of the proposed methods. We also choose the computational steps based on TD-MPC2 paper [2].
>
> [2]Hansen, N., Su, H., & Wang, X. (2024). TD-MPC2: Scalable, Robust World Models for Continuous Control. https://arxiv.org/abs/2310.16828
>
> ---
>
> Q3. There are several typos in the paper. “In BS-MPC, the latent dynamics are modeled using an MLP. We also model the latent dynamics model with an MLP” I believe BS-MPC should be TD-MPC. “we sample M action sets from Gaussian distribution N (μ0, σ0) based on the initial meanμ0 and standard deviation σ0” Missing spacing between mean and \mu_0
>
> A3.  We sincerely thank the reviewer for identifying these typos and providing detailed feedback. We have carefully revised the manuscript to address these issues. We corrected the sentence and simplified the phrasing of the affected sentences to enhance clarity. A revised version of the paper has been uploaded for your review.
>
> ---
>
> Q4. Contribution weakness
>
> A4. We wrote our thoughts in [general reposense](https://openreview.net/forum?id=F07ic7huE3&noteId=x4OIKV1V1Z).

---

> > ### Author Response · Authors · 2024-11-21
> > **Friendly reminder**
> >
> > As the deadline for the rebuttal is in one week, we would like to kindly remind the reviewer to review our response to your comments.
> >
> > If there are any clarifications or additional details we can provide to assist with your review, please do not hesitate to let us know.
> >
> > Thank you for your time and valuable feedback.

---

### Official Review · Reviewer_CXNw · 2024-11-04

**Soundness:** 3
**Presentation:** 3
**Contribution:** 2
**Rating:** 5
**Confidence:** 4

**Summary:**

This paper proposes BS-MPC, a model-based reinforcement learning approach that introduces bisimulation metrics (loss) on top of TD-MPC. Compare to TD-MPC, BS-MPC has an explicit encoder loss term, the adaptation of bisimulation metric, and parallelizing the BS loss. The authors found that their approach can improve training stability, robustness against input noise, and computation efficiency, which is validated on a set of simulation environments.

**Strengths:**

The paper is well-written and easy to follow. The overall presentation is good. The approach is sound and makes sense to the reviewer. The experimental results look promising, compared to TD-MPC.

**Weaknesses:**

However, the major weakness is its novelty.
1. The whole framework is based on TD-MPC. The difference is the authors introduce the Bisimulation metric and its corresponding loss design, which are from the existing literature, as stated in the paper.
2. It is also a common way to introduce additional regularization loss terms for the encoder of model-based RL.
3. The theoretical analysis mainly borrows from the existing work and does not have any major significant result. It would be great if the authors could provide "Under the BS loss training error, what's the performance gap between the final converged policy by their approach and the ideally optimal policy", and "Theoretically, how much performance gain could their approach improve, compared to TD-MPC."

**Questions:**

When you say BS-MPC improves computation efficiency, what does it mean? Is it compared to TD-MPC?
It is surprising to me because BS-MPC has one additional loss term compared to TD-MPC and why is BS-MPC faster to run?

With the above question, I'd like to know the latency overhead of BS loss term in the training.

---

> ### Author Response · Authors · 2024-11-17
> **Author response to reviewer CXNw**
>
> We are grateful for the reviewer’s constructive suggestions and provide our responses below.
>
> Q1. When you say BS-MPC improves computation efficiency, what does it mean? Is it compared to TD-MPC? It is surprising to me because BS-MPC has one additional loss term compared to TD-MPC and why is BS-MPC faster to run?
>
> A1. Yes, BS-MPC achieves faster computational speed than TD-MPC. The primary reason for the speed-up is the structure change in the computation. In BS-MPC, we encode all states s_{t:t+H} at once (z_{t:t+H} = s_{t:t+H}), and hence eliminate the sequential computation. However, TD-MPC needs to compute its latent state z_t  by using the previous step latent state z_{t-1}. Therefore, its calculation flow includes sequential computation, which becomes the bottleneck of TD-MPC. I added a new Appendix in C.2 to give more details about this explanation by using pseudo-code. Also, please refer to Figure 2 for the computational flow difference.
> (Adding bisimulation term does not contribute to the speed-up, but it helps BS-MPC to be robust against the noise in the original state space S)
>
> ---
>
> Q2. With the above question, I'd like to know the latency overhead of BS loss term in the training.
>
> A2. We estimate the calculation cost in Equation (8). Let d denote the dimension of latent space, the Gaussian Wasserstein Distance computational cost is O(d^3). Therefore, the computational cost of the bisimulation metric for Batch size B data becomes O(B d^3).
>
> ---
>
> Q3. Contribution weakness
>
> A3. We wrote our thoughts in [general reposense](https://openreview.net/forum?id=F07ic7huE3&noteId=x4OIKV1V1Z).

---

> > ### Author Response · Authors · 2024-11-21
> > **Friendly reminder**
> >
> > As the deadline for the rebuttal is in one week, we would like to kindly remind the reviewer to review our response to your comments.
> >
> > If there are any clarifications or additional details we can provide to assist with your review, please do not hesitate to let us know.
> >
> > Thank you for your time and valuable feedback.

---

### Official Review · Reviewer_w1co · 2024-11-06

**Soundness:** 3
**Presentation:** 2
**Contribution:** 2
**Rating:** 6
**Confidence:** 3

**Summary:**

This paper presents a new method for model-based reinforcement learning (MBRL) called BS-MPC. The key innovation lies in incorporating a bisimulation metric loss into the objective function to improve encoder stability, robustness to noise, and computational efficiency. By using the bisimulation metric, BS-MPC aims to ensure behavioral equivalence in the latent space, maintaining key characteristics of the original state space. The method is benchmarked against the Temporal Difference Model Predictive Control (TD-MPC) and other model-free and model-based methods on various tasks, showing superior stability and resilience to noise.

**Strengths:**

The paper provides a new perspective by integrating the bisimulation metric to address known challenges in MBRL, particularly around stability and robustness to noise. The experimental results demonstrate how BS-MPC performs well in both state-based and image-based tasks, showing increased resilience to noise and achieving faster training times due to parallel computation. The theoretical analysis adds depth by bounding cumulative rewards in the learned latent space, suggesting that BS-MPC retains meaningful state information effectively.

**Weaknesses:**

While the theoretical foundations are thorough, certain explanations, particularly on encoder stability and noise resilience, could be made clearer to broaden accessibility. The parameters require extensive tuning, which may be impractical for real-world applications lacking automated parameter selection. Additionally, the approach to introducing perturbations, particularly with visual distractions, doesn’t seem entirely effective. It would be beneficial to test perturbations that are more representative of realistic environmental changes, which could better showcase BS-MPC’s resilience.

**Questions:**

Could the authors expand on the sensitivity of BS-MPC to the parameter c4 and potential ways to reduce this dependency?

How does BS-MPC perform in scenarios with dynamic backgrounds that align with the movement instead of pure noise?

Are there additional computational costs associated with bisimulation metric loss, especially in high-dimensional latent spaces?

---

> ### Author Response · Authors · 2024-11-17
> **Author response to reviewer w1co**
>
> We are grateful for the reviewer’s constructive suggestions and provide our responses below.
>
> Q1. Could the authors expand on the sensitivity of BS-MPC to the parameter c4 and potential ways to reduce this dependency?
>
> A1. Thank you for pointing this out. We also think this tuning term is the biggest limitation of the proposed method. In our experiments, we employed grid search for this hyperparameter, which involved six candidates ($10^{-8}$, $0.0001$, $0.001$, $0.01$, $0.1$, $0.5$) making the process computationally manageable. Our findings indicate that smaller values are suitable for high-dimensional and complex tasks, while larger values work better for simpler tasks. Additionally, we believe incorporating techniques such as reward discretization and normalization layers from TD-MPC2 could simplify or even eliminate the need for this tuning in the future.
>
> ---
>
> Q2. How does BS-MPC perform in scenarios with dynamic backgrounds that align with the movement instead of pure noise?
>
> A2. Regrettably, I did not do the experiment in an environment where scenarios with dynamic backgrounds align with the movement. We are setting up the Carla simulator (vehicle simulator) to test the proposed method to see the performance now. We will update the paper as soon as we get the results. However, we believe that BS-MPC is more likely to achieve better performance compared to other methods because the deep bisimulation metric is shown to have good performance in the Carla simulator (dynamic backgrounds that align with the movement)[1].
>
> ---
>
> Q3. Are there additional computational costs associated with bisimulation metric loss, especially in high-dimensional latent spaces?
>
> A3. We estimate the calculation cost in Equation (8). Let d denote the dimension of latent space, the Gaussian Wasserstein Distance computational cost is O(d^3). Therefore, the computational cost of the bisimulation metric for Batch size B data becomes O(B d^3). This is the additional computational cost at each time step compared with TD-MPC. However, since BS-MPC can employ parallel computations, BS-MPC still archives faster calculation time even with this additional computational cost. We add Appendix C.2 to our paper for more details about the parallel computational explanations.
>
> [1]. Zhang, A., McAllister, R., Calandra, R., Gal, Y., & Levine, S. (2021). Learning Invariant Representations for Reinforcement Learning without Reconstruction. https://arxiv.org/abs/2006.10742

---

> ### Author Response · Authors · 2024-11-21
> **Friendly reminder**
>
> As the deadline for the rebuttal is in one week, we would like to kindly remind the reviewer to review our response to your comments.
>
> If there are any clarifications or additional details we can provide to assist with your review, please do not hesitate to let us know.
>
> Thank you for your time and valuable feedback.

---

> ### Author Response · Authors · 2024-11-25
>
> We appreciate your feedback and would like to inform you that we have completed the experiments in scenarios with dynamic backgrounds aligned with movement. Due to the computational demands of the CARLA environment, we were able to run the experiment with only a single seed. The results indicate that BS-MPC outperforms TD-MPC, even in environments where the background image moves along with the car. Additionally, while the BS-MPC loss converges to zero, the TD-MPC loss diverges as training progresses.
>
> We have uploaded the results to the supplemental material. The reviewer can find the relevant figures in the figures folder, specifically named "carla_reward.png" and "carla_consistency_loss.png."

---

### Author Response · Authors · 2024-11-17
**Genenral Response**

First and foremost, I would like to sincerely thank all reviewers for their insightful comments and constructive feedback. They help me critically analyze and enhance my approach. I have replied to each reviewer about their questions and concerns. Here I would like to summarize the changes I made in my manuscript and respond to common questions and concerns raised by multiple reviewers.

# Summary of Paper Revisions
- A new subsection has been added to Appendix A, explaining why BS-MPC outperforms TD-MPC, particularly in noisy environments (as suggested by [seSt](https://openreview.net/forum?id=F07ic7huE3&noteId=N3v9RuT4mp).
- A new subsection has been added to Appendix C (C.2), detailing the computational structure differences between BS-MPC and TD-MPC using pseudo-code. It gives detailed explanations why BS-MPC has faster computational time than TD-MPC.
- Sentences highlighted by [hjrQ](https://openreview.net/forum?id=F07ic7huE3&noteId=21oVCAFWxF) have been corrected and revised to improve clarity and precision.

# Responses to Common Questions
Q1. BS-MPC appears sensitive to hyperparameters. Is there a way to mitigate this issue?

A1. Indeed, tuning the new weight term $c_4$ is essential for achieving optimal performance. In our experiments, we employed grid search for this hyperparameter, which involved six candidates ($10^{-8}$, $0.0001$, $0.001$, $0.01$, $0.1$, $0.5$), making the process computationally manageable. Our findings indicate that smaller values are suitable for high-dimensional and complex tasks, while larger values work better for simpler tasks. Additionally, we believe incorporating techniques such as reward discretization and normalization layers from TD-MPC2 could simplify or even eliminate the need for this tuning in the future.

---

Q2. The contribution appears incremental, as BS-MPC mainly integrates the bisimulation metric into TD-MPC.

A2. We agree that BS-MPC builds upon TD-MPC, and inherits many components from TD-MPC. However, we believe that incremental yet meaningful advancements are essential to driving progress in reinforcement learning and beyond. Many significant contributions in the field, such as TD-MPC2, Dreamer, and Rainbow, have advanced the state of the art by refining and extending existing methods. In this context, we view BS-MPC as a step forward, offering two key contributions:

- Integration of the Bisimulation Metric: While the bisimulation metric has been utilized in some model-based reinforcement learning approaches, this is the first work to incorporate it into a planning-based method. This simple addition provides a novel perspective on how to improve planning robustness and performance. BS-MPC is also the first paper that provides theoretical support for the estimated value.

- Modification of the Computational Flow: BS-MPC revisits and modifies the computational flow used in TD-MPC for calculating costs. Although the values being computed are similar, our changes significantly improve computational efficiency and stabilize the training process compared to TD-MPC.

While these changes may appear incremental, we believe that minimal modifications to existing simple algorithms can play a crucial role in advancing the field and addressing important challenges within the community. Maintaining the simplicity of TD-MPC while addressing three of its open problems demonstrates the importance of targeted refinements to simple yet powerful algorithms. We hope this contribution will inspire further exploration in the community.

---

Q3. Are there additional computational costs associated with bisimulation metric loss, especially in high-dimensional latent spaces?

A3. We estimate the calculation cost in Equation (8). Let d denote the dimension of latent space, the Gaussian Wasserstein Distance computational cost is O(d^3). Therefore, the computational cost of the bisimulation metric for Batch size B data becomes O(B d^3). This is the additional computational cost at each time step compared with TD-MPC. However, since BS-MPC can employ parallel computations, BS-MPC still archives faster calculation time even with this additional computational cost. We add Appendix C.2 to our paper for more details about the parallel computational explanations.

---

### Author Response · Authors · 2024-11-20
**Paper Update**

As suggested by the reviewer [seSt](https://openreview.net/forum?id=F07ic7huE3&noteId=N3v9RuT4mp), we have conducted the additional experiments for TD-MPC2 and updated the PDF with the new results. In the updated PDF, we have moved Appendix D to Section 5 to better highlight the experimental results, specifically by including the TD-MPC2 results in the main body of the paper.

We have finished responding to all of the comments we got from the reviewers, and we are looking forward to having discussions with each reviewer.

---

### Author Response · Authors · 2024-11-25
**Paper Update2**

We added a new experimental result in Carla simulator suggested by reviewer [w1co](https://openreview.net/forum?id=F07ic7huE3&noteId=ggQrsRBRSf).

## Summary of the new experimental result
BS-MPC outperforms TD-MPC, even in environments where the background image moves along with the car. Additionally, while the BS-MPC loss converges to zero, the TD-MPC loss diverges as training progresses.

## Where to find the results
We have uploaded the results to the supplemental material. The reviewer can find the relevant figures in the figures folder, specifically named "carla_reward.png" and "carla_consistency_loss.png."

---

### Meta-Review · Area_Chair_npgw · 2024-12-21

**Metareview:**

The paper presents Bisimulation Metric for Model Predictive Control (BS-MPC), a technique for model-based reinforcement learning that uses a bi-simulation metric to improve encoder stability, robustness to noise, and computational efficiency. The benefits of BS-MPC are demonstrated via experiments on both continuous control and image-based tasks from the DeepMind Control Suite.

**Additional Comments On Reviewer Discussion:**

The authors provided detailed responses to the concerns raised by the reviewers. Even after the discussion period the reviewers remained  split on whether the paper meets the bar for acceptance. Unfortunately, reviewer CXNw did not engage with the authors or other reviewers during the rebuttal and discussion despite recommending rejection. To be fair to the authors I am not taking CXNw's rejection recommendation into consideration for the final decision.
Reviewer seSt provided a dissenting opinion about unresolved issues regarding the claim of robustness in the paper. I thank the reviewer for engaging with the authors and providing suggestions for improving the paper. The key disagreement seems to be whether the claim of robustness over TD-MPC can be made without comparing the respective Lipschitz constants. Having carefully reviewed the discussion and the paper, I feel that the authors have provided sufficient empirical evidence to justify their claim of robustness. In particular, the paper does not claim to show that BS-MPC is provably more robust than TD-MPC, but only that experimental results show that it is more robust. While a theoretical guarantee of improved robustness would certainly improve the paper, it is currently unknown if such a guarantee exists and hence empirical evidence of robustness should be considered sufficient in view of the other scientific contributions of this paper. As it stands, especially accounting for the changes already incorporated in the paper after the discussion period, the paper makes a significant enough contribution to warrant acceptance.

---

### Decision · Program_Chairs · 2025-01-22

Accept (Poster)